# Blood–Nerve Barrier (BNB) Pathology in Diabetic Peripheral Neuropathy and In Vitro Human BNB Model

**DOI:** 10.3390/ijms22010062

**Published:** 2020-12-23

**Authors:** Yukio Takeshita, Ryota Sato, Takashi Kanda

**Affiliations:** Department of Neurology and Clinical Neuroscience, Graduate School of Medicine, Yamaguchi University, Yamaguchi 753-8511, Japan; takeshy@yamaguchi-u.ac.jp (Y.T.); ryota214@yamaguchi-u.ac.jp (R.S.)

**Keywords:** blood–nerve barrier, pericyte, Schwann cells, advanced glycation end products

## Abstract

In diabetic peripheral neuropathy (DPN), metabolic disorder by hyperglycemia progresses in peripheral nerves. In addition to the direct damage to peripheral neural axons, the homeostatic mechanism of peripheral nerves is disrupted by dysfunction of the blood–nerve barrier (BNB) and Schwann cells. The disruption of the BNB, which is a crucial factor in DPN development and exacerbation, causes axonal degeneration via various pathways. Although many reports revealed that hyperglycemia and other important factors, such as dyslipidemia-induced dysfunction of Schwann cells, contributed to DPN, the molecular mechanisms underlying BNB disruption have not been sufficiently elucidated, mainly because of the lack of in vitro studies owing to difficulties in establishing human cell lines from vascular endothelial cells and pericytes that form the BNB. We have developed, for the first time, temperature-sensitive immortalized cell lines of vascular endothelial cells and pericytes originating from the BNB of human sciatic nerves, and we have elucidated the disruption to the BNB mainly in response to advanced glycation end products in DPN. Recently, we succeeded in developing an in vitro BNB model to reflect the anatomical characteristics of the BNB using cell sheet engineering, and we established immortalized cell lines originating from the human BNB. In this article, we review the pathologic evidence of the pathology of DPN in terms of BNB disruption, and we introduce the current in vitro BNB models.

## 1. Introduction

Diabetes mellitus, especially type 2 diabetes mellitus, affects approximately 463 million people worldwide and is expected to affect more than 700 million people by 2045 [1]. Diabetic peripheral neuropathy (DPN) is one of the most common complications of diabetes mellitus and occurs in 30%–50% of patients with type 2 diabetes mellitus [2]. DPN is considered to occur in 100% of patients with type 1 diabetes mellitus within a disease duration of 15 years and in 30% of patients with type 2 diabetes mellitus within a disease duration of 25 years [3]. As overcoming DPN is an important global challenge, studies have been conducted on the pathology of DPN and have proposed various pathological hypotheses.

In peripheral nerve tissues, where glucose is taken up by non-insulin-dependent glucose transporters, most of the glucose is transported from blood vessels to peripheral nerve tissues, according to the concentration gradient. Hyperglycemia directly contributes to the elevation of intracellular glucose concentrations. In a hyperglycemic state, the polyol metabolic pathway is enhanced via increased activity of aldose reductase in peripheral nerve tissues [4]. Edema, degeneration, and ischemia of nerves are considered to occur via various mechanisms, such as osmotic stress, abnormal protein kinase C activity, increased oxidative stress, accumulation of advanced glycation end products (AGEs) due to impaired glucose metabolism, and decreased production of nitric oxide [5]. Notably, disruption of the blood–nerve barrier (BNB), a strict barrier system that stabilizes the internal environment of peripheral nerves, has been increasingly reported to be deeply involved in the onset and exacerbation of DPN with in vitro BNB models. Here, we present recent findings on the pathology of DPN in terms of BNB disruption and recommend our current in vitro BNB models.

## 2. Anatomy of the BNB and Peripheral Nerves

Peripheral nerve tissues mainly compose the BNB, Schwann cells, and nerve axons. The BNB is mainly localized in the microvessels of peripheral nerves within the peripheral neural parenchyma (endoneurium) and consists of vascular endothelial cells bound by tight junction proteins, pericytes attached to the outer side of the vascular endothelial cells, and basal laminae covering these two types of cells. Like the blood–brain barrier (BBB) in the central nervous system, the BNB acts as a wall preventing the influx of harmful substances circulating in the blood into peripheral nerves [6]. The BBB is mainly composed of brain microvascular endothelial cells, pericytes, astrocytes, and the two distinct basement membranes. The BNB has a similar structure to the BBB, with the exception of lacking astrocytes (Figure 1A). It had previously been thought that the BNB was leaky compared to the BBB, as a result of the lack of astrocytes, but several reports showed that the BNB has a barrier function as robust as that of the BBB [7,8] and acts as an interface to maintain the microenvironment in peripheral nerves by expressing transporters for the transportation of nutrients, such as glucose and amino acids, as well as for the excretion of waste products [6,9]. The basement membrane in the BBB and BNB consists of a mixture of extracellular matrix proteins including collagen IV and laminin [10]. Laminin is a trimeric molecule comprising α-, β-, and γ-subunits and shows specific expression in the central and peripheral nervous systems [10]. We showed that laminin α4 was hardly expressed in the BNB compared with the BBB (Figure 1B), whereas α5, β1, β2, and γ1 were widely expressed in the BBB and BNB [11]. In experimental autoimmune encephalomyelitis, a relevant preclinical model of multiple sclerosis laminin α4 facilitates the transmigration of T lymphocytes across the BBB because integrin α6β1, a major counterpart receptor of laminin α4, is strongly expressed on all T lymphocytes [12]. These results showed that T lymphocytes, with integrin α6β1, migrated across the BNB through other factors (except for laminin α4 in peripheral autoimmune neuropathies), suggesting the BNB has its own barrier system because of a lack of laminin α4 in the BNB. On the other hand, laminin α5 plays a key role in the maintenance and repair of BBB after hypoxic injury and inflammation [13]. Therefore, we hypothesized that BNB-specific laminin expression without laminin α4 might promote almost the same properties as a barrier system as those with the BBB.

In contrast, Schwann cells surround the axons of neurons to form a myelin sheath, a structure responsible for saltatory conduction. Together with pericytes, Schwann cells are involved in providing nutrients to neurons and play a role in maintaining the microenvironment [14].

### 2.1. Establishment of BNB-Forming Cell Lines and Functions of Pericytes

#### 2.1.1. Establishment of BNB-Forming Cell Lines

To clarify the mechanism underlying BNB disruption in DPN, we established temperature-sensitive immortalized cell lines of vascular endothelial cells and pericytes originating from human BNB (Figure 2A,B). We extracted endoneurial vessels from human sciatic nerves obtained at autopsy. Endothelial cells and pericytes were then isolated and cultured. These cells were immortalized by introducing the temperature-sensitive SV40 large T-antigen gene using the same retrovirus vector. The vascular endothelial cells were characterized by their spindle-fiber-shaped morphology, expression of von Willebrand factor, expression of tight junction-related molecules, and high electrical resistance, which were consistent with the characteristics of BNB-forming endothelial cells [7,15]. Pericytes were characterized by their cobblestone-shaped morphology and identified by their ruffled borders and the expressions of α-smooth muscle actin (α-SMA), PDGF-β receptor (PDGFβR), osteopontin, and desmin [16,17]. After our report, Vosef et al. established BNB-forming vascular endothelial cell lines derived from human sciatic nerves [18,19]. These cell lines were immortalized by introducing the SV40 large T-antigen gene. However, the continuous passage of immortalized cell lines by serial introduction of the SV40 large T-antigen gene may alter the morphology of cells, leading to a loss of their biochemical function [20]. The human cell lines derived from endothelial cells forming the BNB (hEC-BNB) and the pericytes forming the BNB (hPCT-BNB), established in the present study, were conditionally immortalized by introducing the temperature-sensitive SV40 large T-antigen gene (Figure 2C). The SV40 large T-antigen gene was inactivated by culturing at 37 °C, leading to a loss of immortalization. The cell lines were, therefore, considered to have physiological and biochemical functions similar to in vivo conditions [7,17]. 

#### 2.1.2. Function of Pericytes in the BNB

In the BBB, endothelial cells function as an important barrier; however, they alone cannot maintain the physiological barrier, and continuous “crosstalk” between cellular and non-cellular components surrounding these cells is considered essential for maintaining the barrier. Astrocytes have been recognized as important BBB regulators [21], and in recent years, pericytes have also been described as important BBB regulators [22,23]. In 2010, Armulik et al. reported transcytosis in BBB-forming endothelial cells, leading to increased permeability of the BBB in pericyte-deficient mice [16]. Since cells such as astrocytes are not present in the BNB, it is assumed that pericytes play an important role in regulating the BNB. Using BNB-forming cell lines, we previously demonstrated the following results: (1) pericyte culture supernatant increased the electrical resistance (trans-endothelial electrical resistance) and decreased the permeability of 14C-labeled inulin in endothelial cell lines originating from the BNB; (2) pericyte culture supernatant increased the expression of tight junction-related proteins (claudin-5, occludin) in endothelial cell lines originating from the BNB; (3) pericytes secrete physiologically active substances (angiopoietin-1 (Ang-1), vascular endothelial growth factor (VEGF), basic fibroblast growth factor (bFGF), and a neurotropic factor (glial cell line-derived neurotrophic factor, GDNF)); (4) Ang-1 and VEGF secreted by pericytes decreased the expression of claudin-5, leading to a tendency to decrease BNB function; and (5) bFGF and GDNF, in contrast, increased the expression of claudin-5 [17,24]. These findings demonstrate that pericytes play an important role as BNB regulators.

#### 2.1.3. Construction of a New In Vitro BNB Model

BNB plays crucial roles in the maintenance of peripheral nervous system homeostasis and in the pathogenesis of DPN. The BNB components (pericytes and basement membrane such as laminin α5, β1, γ1) have directly contact with the endothelial cells and participate the regulation of each barrier function. However, the mechanism of this regulation still remains unknown because of the non-existence of the in vitro human BNB model in which these components can directly come in contact with endothelial cells (Figure 3A,B). Recently, we constructed a new in vitro human BNB model that incorporated a multi-culturing system of these BNB components using the UpCell technology of Nunc (Figure 3C). In this model, firstly, hPCT-BNBs were cultured on the luminal side of the insert-membrane (Figure 3D). Secondly, hEC-BNBs were cultured in an UpCell dish with a coated temperature-responsive polymer, which can achieve sheet-like detachment of confluent cells and extracellular matrix by temperature-shifting to 20 °C. Then, sheet-like detachment of confluent hEC-BNBs were transferred onto the hPCT-BNBs. Confocal 3D analysis with live staining of each cell line showed that the multi-cultured insert constituted the two-layer structures, which consisted of hEC-BNBs and hPCT-BNBs. The layer of hPCT-BNBs was in direct contact with the hEC-BNBs (Figure 3E).

## 3. BNB Disruption

### 3.1. Impaired Barrier Function Owing to Advanced Glycation End Products (AGEs)

The association between diabetes mellitus and barrier function has been investigated for a long time, and impaired barrier function is considered to be the first stage in the development of DPN. In 1987, a study demonstrated that patients with type 1 DPN had increased vascular permeability and high concentrations of high-molecular-weight proteins, such as albumin and immunoglobulin G (IgG), in the endoneurium. This study revealed, for the first time, that barrier function is impaired by BNB disruption in DPN [25]. In the following year, another study demonstrated that diabetes patients without comorbid peripheral neuropathy also had increased vascular permeability and high concentrations of high-molecular-weight proteins, such as albumin and IgG, in the endoneurium. It was confirmed that chronic hyperglycemia, per se, impairs the barrier function of the BNB [26]. Moreover, it has been reported that when high-molecular-weight proteins transferred to the endoneurium accumulate over a long time, electrolyte regulation is impaired in peripheral nerves, and impaired electrolyte regulation leads to the progression of edema, which causes ischemia of the nerve fascicles [27]. Furthermore, these high-molecular-weight proteins transferred to the endoneurium also include neuropathic humoral factors, which are important risk factors for not only DPN but also various peripheral neuropathies.

Advanced glycation end products (AGEs) are neurotoxic humoral factors released into the endoneurium. AGEs are rapidly generated by a continuous hyperglycemic state and are greatly involved in the pathogenesis of diabetic microvascular complications, including nephropathy, retinopathy, and dementia [28,29]. In diabetic neuropathy, AGEs accumulate in Schwan cells and axons in peripheral nerves and in BNB-forming cells (vascular endothelial cells, pericytes, and basement membrane) [30]. AGEs are also involved in the progression of diabetic neuropathy. In diabetic nephropathy, glomerular basement membrane thickening occurs. The mechanism involves the release of TGF-β from AGE-stimulated mesangial cells, which increases the production of type IV collagen that forms the basement membrane [31]. In diabetic retinopathy, AGEs accumulate in retinal vessels, which increase VEGF expression, leading to the disruption of the blood–retinal barrier and an increase in the permeability of retinal microvascular vessels [32].

Our study using immortalized cell lines of endothelial cells and pericytes originating from human BNB showed that AGEs decreased the amount of claudin-5 by increasing the autocrine secretion of VEGF from endothelial cells forming the BNB, leading to disrupted barrier function, thereby suggesting that AGEs are closely involved in the pathogenesis of BNB disruption (Figure 4).

### 3.2. Microvascular Basement Membrane Thickening in DPN

In DPN, as described above, hyperglycemia directly causes cellular dysfunction in peripheral nerve tissues, such as enhancement of the polyol metabolic pathway. Furthermore, BNB disruption is induced, resulting in various pathological conditions including impaired barrier function, induction of local inflammation, poor circulation, hypoxia, and damage to Schwann cells and neurons.

Since the late 1970s, studies have reported morphological changes of BNB that suggest its disruption in DPN. These changes include hyperplasia of endothelial cells, dissociation of the tight junction of endothelial cells, microvascular occlusion, fenestrated microvascular endothelial cells, endoneurial microvascular basement membrane thickening, and degeneration of microvessel-forming endothelial cells and pericytes [33,34,35]. Among these, microvascular basement membrane thickening (Figure 5) is a common finding in diabetic neuropathy; however, its molecular mechanisms have not been elucidated.

Using immortalized cell lines of endothelial cells and pericytes originating from human BNB, we showed that pericytes played an important role in producing proteins forming the basement membrane, such as fibronectin and type IV collagen, and tissue inhibitor of metalloprotease-1 (TIMP-1), a molecule inhibiting the degradation of the basement membrane. Thus, pericytes play a crucial role in maintaining the basement membrane in the BNB (Figure 6).

In a hyperglycemic state, AGEs accumulate in pericytes and increase the autocrine secretion of VEGF and TGF-β, leading to the enhanced production of fibronectin and type IV collagen, suggesting that microvascular basement membrane thickening in diabetic neuropathy is caused by the accumulation of AGEs in pericytes under hyperglycemic conditions via VEGF and TGF-β signaling by autocrine secretion by pericytes (Figure 6) [36,37].

### 3.3. Induction of Local Inflammation

Although DPN is not usually classified as an inflammatory neuropathy, several phenomena are known to occur in diabetic nephropathy and retinopathy. In the microvessels, the endothelium is damaged by oxidative stress, hypoxia, and metabolites such as AGE (bound to the receptors for AGE); consequently, multiple inflammatory signaling cascades are stimulated, and proinflammatory cytokines, such as interleukin 1β and tumor necrosis factor-α, induce local inflammation [38]. Furthermore, the expression of genes associated with inflammation and immune responses was enhanced in a mouse model of type 2 diabetes mellitus [39,40]. It has been hypothesized that DPN induces local inflammation, which in turn contributes to the onset of disordered immune responses and cell infiltration [41]. Moreover, elevated expression of genes associated with phagosome formation and excessive stimulation of the toll-like receptor signaling pathway in macrophages in nerve tissues have been reported in patients with DPN [42]. A recent study on sural nerve biopsy in diabetes patients revealed elevated expression of CD40, a surface antigen affecting inflammation and thrombosis in vascular endothelial cells [43]. In 2020, Gonçalves et al. reported that neurotrophin receptor p75, which is expressed by the Schwann cells, was an important key molecule to activate the immune-related pathways and lysosomal stress with in vivo analysis [44]. These findings provide important evidence for the pathology of local inflammatory cell infiltration and microvasculitis in DPN.

### 3.4. Disorders Due to Poor Circulation and Hypoxia

In DPN patients, pathological examination of sural nerve samples shows abnormal vascular findings characterized by swollen vascular endothelial cells in small blood vessels in the endoneurium forming the BNB and a narrowed vascular lumen due to the overlapping and thickening of multiple basal laminae (Figure 5D). Such abnormal blood vessels not only show a decreased expression of tight junction-associated proteins in the blood vessels, but they also show proliferation of the vascular wall due to fibrin, and loss of pericytes surrounding the vascular endothelium. These findings indicate BNB disruption. It is known that the loss of nerve fibers is generally homogeneous across nerve fascicles, which reflects the metabolic disorder due to hyperglycemia (Figure 5B). However, cases have been reported in which the density of residual myelinated nerves was inhomogeneous across nerve fascicles as well as within the same nerve fascicle [45]. These cases indicate that the pathology of DPN includes ischemic neuronal damage due to advanced narrowing of the vascular lumen in addition to metabolic disorder of the peripheral nerves due to hyperglycemia. In fact, decreased oxygen partial pressure in the endoneurium has been previously reported in patients with DPN [46,47,48,49]. This explains the ischemic condition of peripheral nerve tissues due to poor circulation. In a mice model, it has been shown that tissue ischemia stimulates macrophages to release vascular endothelial growth factors (VEGFs) [50,51]. As VEGFs increase vascular permeability in the BNB [17], they appear to be important factors for BNB disruption in DPN. Moreover, in the peripheral nerve tissues of patients with DPN, the expression of the transcription factor hypoxia inducible factor-1 (HIF-1), which is induced by an ischemic state, is elevated [43,52]; this induces the expression of various downstream molecules for cellular repair. In particular, the expressions of phosphatase and tensin homolog, which is important for axon repair, and CD40, a surface protein of vascular endothelial cells important for cell infiltration, are elevated [53,54]. In contrast, HIF-1 increases the expression of nicotinamide adenine dinucleotide phosphate oxidase, which induces the production of cytotoxic reactive oxygen species (ROS) in the vascular wall [55,56]. Similar to the pathology reported in the central nervous system [57], the ROS generated in the vascular wall is assumed to cause oxidative damage to the BNB, destruction of the tight junction, and tissue destruction by the activation of matrix metalloproteinases, thereby disrupting the BNB.

## 4. The Association between Disruption of Schwann Cells and BNB

Schwann cells are the most abundant cells in peripheral nerve tissues. The cells surround the axons of neurons to form myelinated and unmyelinated nerves. Owing to close interaction with axons and BNB cells, Schwann cells maintain the homeostasis of peripheral nerve tissues [8]. When metabolic disorder due to hyperglycemia occurs in Schwann cells, peripheral neuropathies, such as demyelination due to the destruction of myelin, disruption of homeostasis in axons, and impaired axonal regeneration, are induced [8]. As described above, a hyperglycemic state enhances the polyol metabolic pathway via increased activity of aldose reductase, even in Schwann cells [58,59,60], and induces the production of cytotoxic ROS. The generated ROS cause oxidative damage and mitochondrial dysfunction, which develop into cellular disorders of Schwann cells and axons [61]. In a hyperglycemic state, Schwann cells produce the chemokines CXCL9, CXCL10, and CXCL11, which induce the local infiltration of CD8+ T-cells [62]. This infiltration leads to the apoptosis of Schwann cells. Consequently, axons are further damaged [62]. Although it is well known that the disruption of Schwan cells by hyperglycemia induced DPN, the interaction between BNB and Schwann cells in DPN is still unknown because of the lack of adequate in vitro human BNB models with Schwann cells and peripheral nerve.

## 5. Etiology of Conditions Other than Hyperglycemia

Needless to say, hyperglycemia is an important factor for DPN. Recent epidemiological studies on DPN in patients with type 1 and type 2 diabetes mellitus have indicated coronary vascular risk factors, such as obesity, dyslipidemia, hypertension, and smoking, might be also involved in the onset of DPN, especially in type 2 diabetes mellitus [63,64]. Treatment of hyperglycemia is reported to reduce the incidence of DPN by approximately 60%–70% in patients with type 1 diabetes mellitus [65,66] but by only approximately 5%–7% in patients with type 2 diabetes mellitus [67,68]. Furthermore, at least 40% of patients with type 2 diabetes mellitus develop DPN despite favorable glycemic control [69,70]. As many detailed aspects of molecular mechanisms of these risk factors are unknown, it is possible that these factors exert a bad influence on the BNB.

## 6. Conclusions

As described above, DPN is associated with axonal degeneration caused by metabolic disorder in each type of cell in peripheral nerve tissue due to hyperglycemia and by disruption of the homeostatic mechanism of peripheral nerves due to dysfunction of the BNB and Schwann cells (Figure 7). Previous studies have demonstrated the disruption of the BNB in DPN. Although many pathological findings were found in studies conducted up to around 1980, the molecular mechanism has not been completely investigated owing to limitations in in vitro analysis. As previous epidemiological studies have reported hyperglycemia and coronary vascular risk factors, such as obesity, dyslipidemia, hypertension, and smoking, in DPN, there is a need for research on DPN with a perspective other than hyperglycemia. We succeeded in developing an in vitro BNB model to reflect the anatomical characteristics of the BNB using cell sheet engineering and newly established immortalized cell lines originating from human BNB (Figure 2). We expect that the new in vitro model offers a potential breakthrough for molecular biology studies of DPN in the BNB.

## Figures and Tables

**Figure 1 ijms-22-00062-f001:**
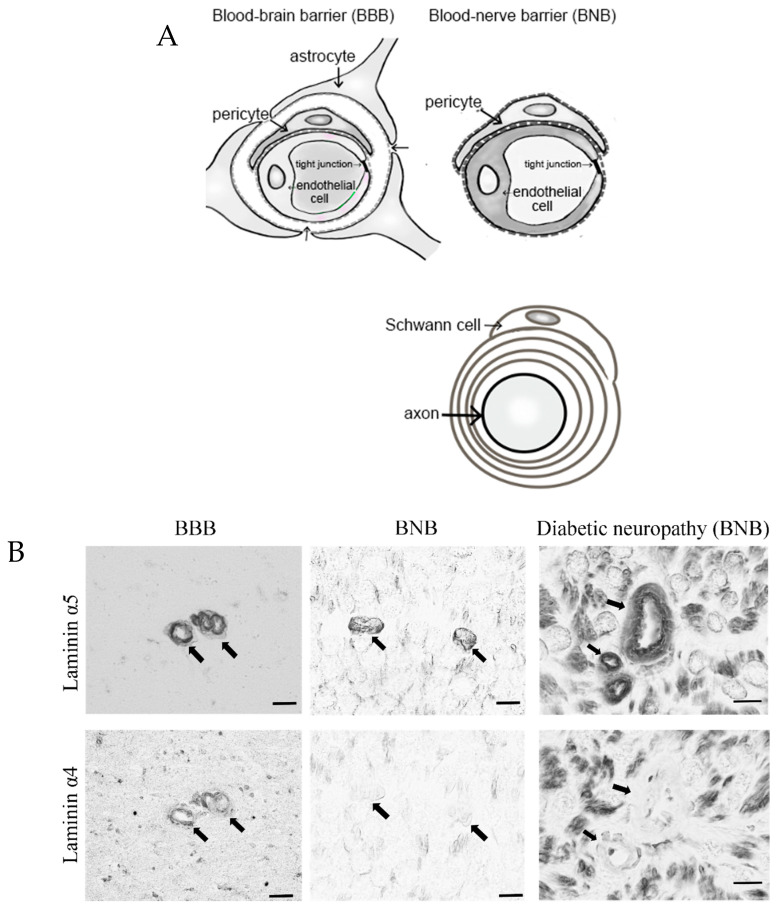
Comparative anatomy of the blood–brain barrier and blood–nerve barrier. (**A**) Peripheral nerve tissue consists of the blood–nerve barrier and axons of neurons wrapped in Schwann cells. Unlike the blood–brain barrier, which consists of vascular endothelial cells, pericytes, astrocytes, and basal laminae, the blood–nerve barrier consists of vascular endothelial cells, pericytes, and basal laminae. (**B**) Immunostaining with anti-laminin (α5 and 4) in the blood–brain barrier (BBB; brain; left panel) and blood–nerve barrier (BNB; median nerve; middle panel) from a postmortem 67-year-old man with amyotrophic lateral sclerosis and from a 67-year-old man with diabetic peripheral neuropathy (DPN; sural nerves; right panel). Laminin α5 immunoreactivity was localized to the microvessels. Only laminin α4 immunoreactivity was lost in the BNB. Bar 10 μm.

**Figure 2 ijms-22-00062-f002:**
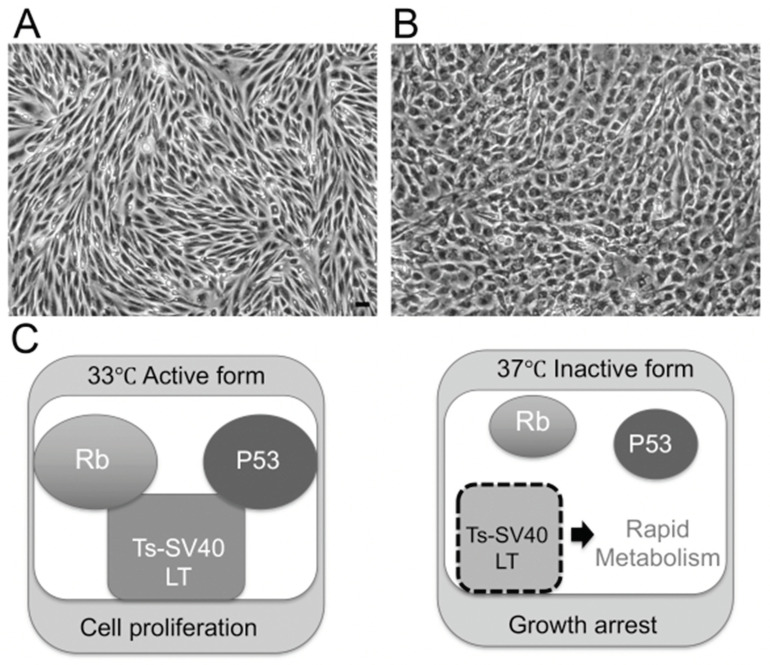
Establishing the BNB cell lines that maintain BNB properties. (**A**) Morphology of hEC-BNB (human endothelial cells in BNB) is spindle-shaped. (**B**) Morphology of hPCT-BNB (human pericytes in BNB) is cobblestone-shaped. (**C**) Temperature-sensitive SV40 large T antigen. At 33 °C, TS-SV40-LT binds and inhibits p53 and Rb, which are strong tumor suppressors, leading to continuous cell proliferation. At 37 °C, TS-SV40-LT is inactivated, and the cells exhibit growth arrest and differentiate into endothelial cells. Scale bar represents 10 µm.

**Figure 3 ijms-22-00062-f003:**
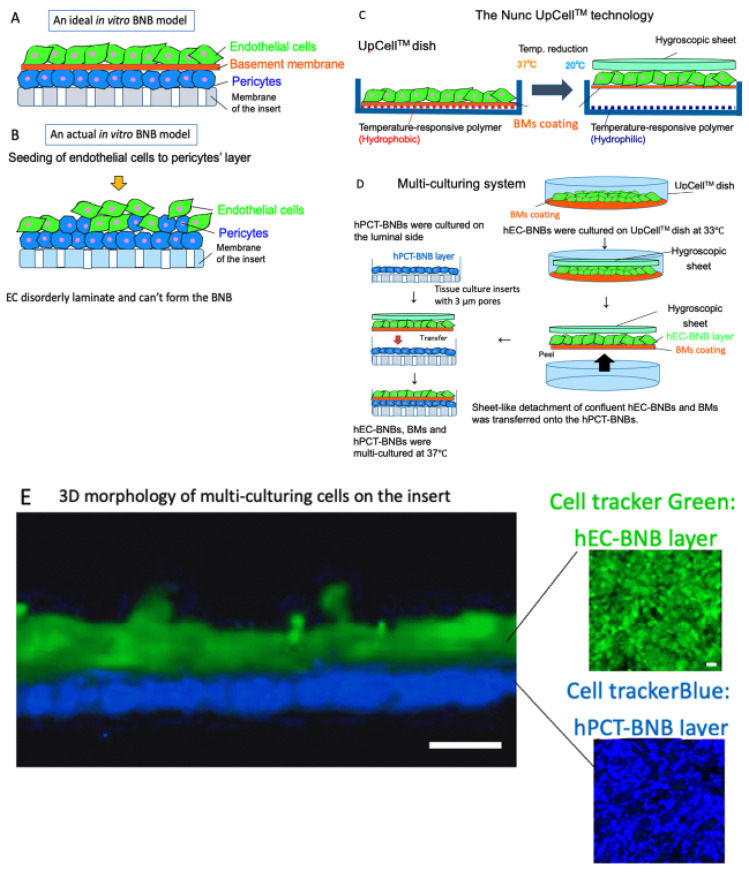
Construction of a new in vitro BNB model. (**A**) An ideal in vitro human BNB model. The endothelial cells form the monolayer. The BNB components (pericytes and basement membrane) are in direct contact with the monolayer of endothelial cells on the tissue culture insert. (**B**) An actual in vitro human BNB model. If endothelial cells are forcibly multi-cultured on the layer of pericytes, they disorderly laminate and cannot form the BNB due to the difference in growth speed of each cell line. (**C**) The Nunc UpCell^TM^ technology. Temperature-responsive polymer is immobilized on the surface of the UpCell^TM^ dish. The polymer-grafted surface shows reversible hydrophobic and hydrophilic properties across the threshold temperature of 20 °C. The cell-sheet is detached from the dish without harmful enzymes (e.g., trypsin or dispase) and attached to the hygroscopic sheet to transfer. (**D**) Multi-culturing systems. hPCT-BNBs were cultured on the luminal side of the cell culture insert membrane with 3µm pores. After flipping the cultured insert, hPCT-BNBs were cultured on the luminal side. hEC-BNBs were cultured in an UpCell dish. After incubation at 20 °C, the sheet of confluent hECs was detached and transferred onto the hPCT-BNBs. (**E**) Confocal 3D analysis with live staining of hEC-BNBs and hPCT-BNBs. Multi-cultured insert constituted the two-layer structures, which consisted of hEC-BNBs and hPCT-BNBs. The layer of hPCT-BNBs was close to the hEC-BNBs. Bar = 5 µm.

**Figure 4 ijms-22-00062-f004:**
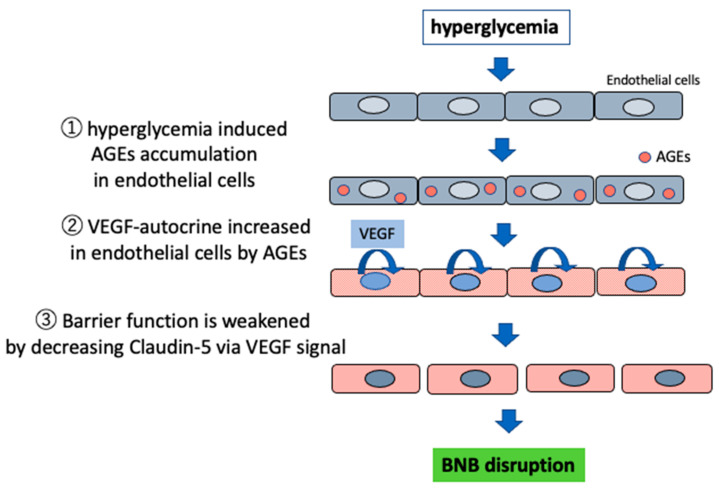
BNB disruption in hyperglycemia by advanced glycation end products (AGEs). Hyperglycemia induced AGE accumulation in endothelial cells. Then, accumulated AGEs increased autocrine VEGF in endothelial cells. Finally, VEGF signal decreased claudin-5 and weakened barrier function.

**Figure 5 ijms-22-00062-f005:**
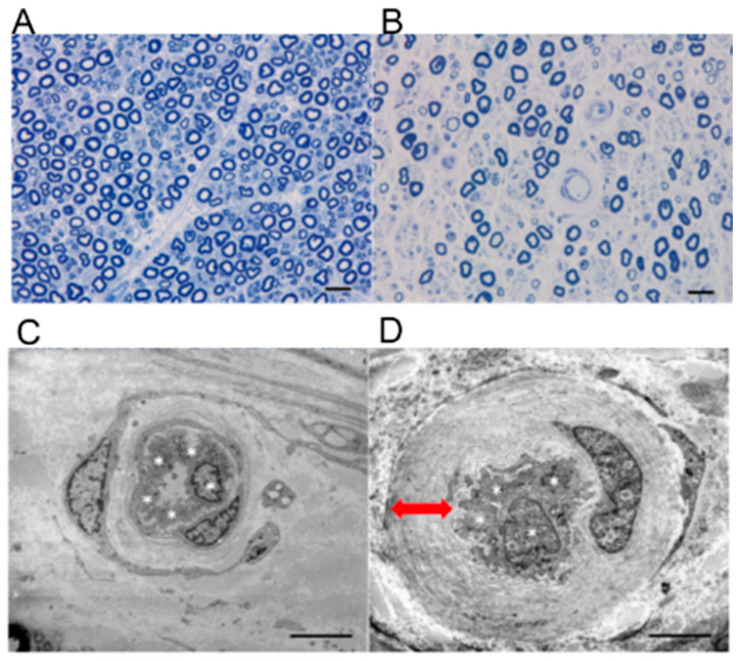
Pathological findings of diabetic peripheral neuropathy. Upper row: Images of sural nerve biopsy specimens from a patient with diabetic peripheral neuropathy and a healthy individual (Epon-embedded specimens; toluidine blue staining); (**A**) healthy case, 55 year old male; (**B**) diabetic peripheral neuropathy case, 60 year old male. Compared with the specimen from the healthy individual, the specimen from patient with diabetic peripheral neuropathy shows more extensive loss of myelinated nerve fibers in the endoneurium. Generally, residual myelinated nerve fibers are homogeneously distributed in the same nerve fascicle. Bar = 50 μm. Lower row: Electron micrographs of small blood vessels in the endoneurium (left, healthy case; right, diabetic peripheral neuropathy case). (**C**) The cells in the specimen from a patient with diabetic peripheral neuropathy are swollen such that they almost occlude the lumen of the vessel. (**D**) Compared with those in the specimen from the diabetic peripheral neuropathy case, in the specimen from a patient with diabetic peripheral neuropathy, the small blood vessel is composed of four vascular endothelial cells (*). Multiple basal laminae overlap and are thickened (↔). Bar = 2 μm.

**Figure 6 ijms-22-00062-f006:**
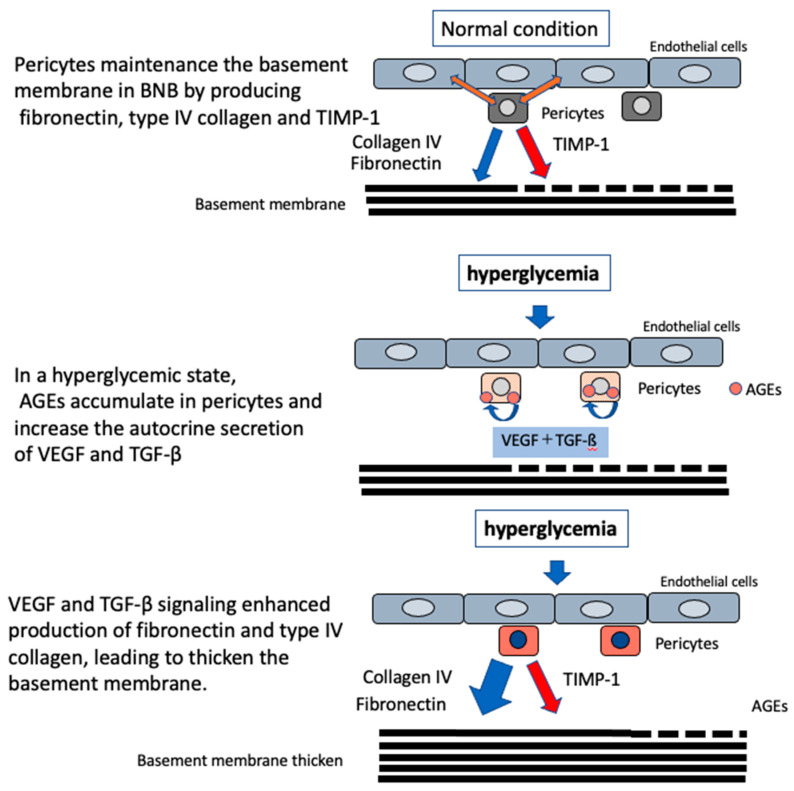
Molecular mechanism modeling of basement membrane thickening in diabetic neuropathy. In normal condition, pericytes maintain the basement membrane in the BNB by producing fibronectin, type IV collagen, and TIMP-1. In hyperglycemia, AGEs accumulate and increase the autocrine secretion of VEGF and TGF-β in pericytes. VEGF and TGF-β signaling enhanced production of fibronectin and type IV collagen, leading to thickening the basement membrane.

**Figure 7 ijms-22-00062-f007:**
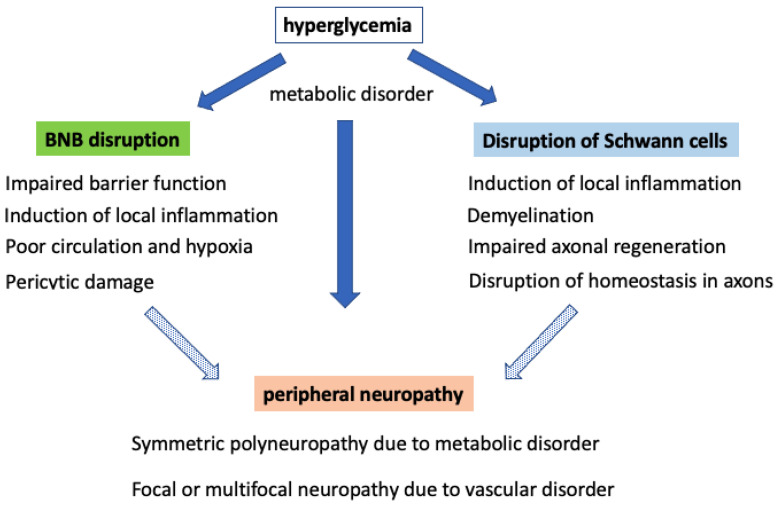
The pathological mechanism of neuropathy in diabetes mellitus. Hyperglycemia affects the BNB, Schwann cells, and peripheral nerves. BNB disruption has been confirmed to be associated with various pathological conditions: (1) impaired barrier function by accumulation of cytotoxic high-molecular-weight proteins, (2) induction of vasculitis and the progression of neuropathy due to induced local inflammation, (3) progression of neuropathy due to poor circulation and hypoxia, and (4) disruption of the regulatory mechanism of the BNB due to damaged pericytes. In Schwann cells, metabolic disorder induces the development of peripheral neuropathies such as demyelination, disruption of axonal homeostasis, and impaired axonal regeneration due to the destruction of myelin. Finally, disruption of the BNB and Schwann cells induced clinical types of peripheral neuropathy: (1) symmetric polyneuropathy due to metabolic disorder and (2) focal or multifocal neuropathy due to vascular disorder.

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
