# Peer review of "Blood–Nerve Barrier (BNB) Pathology in Diabetic Peripheral Neuropathy and In Vitro Human BNB Model"

_ijms, 2020, doi:10.3390/ijms22010062_

Round 1

Reviewer 1 Report

This paper attempted to review the recent literature on the effect of blood-nerve-barrier (BNB) disruption in diabetic peripheral neuropathy (DPN) while also reported characterization of in-vitro cell-lines of vascular endothelial cells and pericytes originated from BNB of human sciatic nerve. They briefly described anatomy of BNB and how they established the in-vitro cell lines from BNB and then gave account on key mechanisms of BNB disruptions that occur in DPN in human as well as animal models of diabetes.

The cell lines from human sciatic nerve described in this study will be an essential tool in further understanding BNB functions in DPN and will assist in developing therapeutic strategy targeting BNB functions. 

However I have following comments/suggestions: 

General comments:

  1. Introduction need to be better phrased with more detailed (one paragraph) about what happens to BNB in healthy and diseased/diabetic conditions.
  2. Conclusion was almost same as introduction which need to stand alone, phrased differently and with more future perspectives for a review article. 
  3. English need to be improved in many places for clarity and to avoid misleading information. 

Specific comments:

  1. Abstract- Page 1, line 9: the sentence says "axons of Schwann cells" which is bizarre, how can Schwann cells have axons?
  2. Introduction- Page 1, line 34-35: This reference is incorrect and old. Need a more recent reference (e.g.; PMID: 31518657)
  3. Page 2, line 45-46: the sentence says: ..."peripheral nerve tissues, such as retina and kidneys"  retina and kidneys are not peripheral nerve tissues. 
  4. References are missing in several places: Page 2, line 47-49 (In a hyper....  tissues). Page 2, line 61-63 (Like the ......   peripheral nerves). Page 2, end of line 68. Page 9, line 257-258 (Owing to ...   nerve tissues). Page 9, line 28-261 (When ......are induced). Page 9, line 264-266 (In a hyperglycemic.. CD*+ T-cells).
  5. Figure 1: Title should be " comparative anatomy of BBB and BNB"
  6. Figure 2: fix English in title. Hard to distinguish panel A and B. Is there a colored image? What was the scale bar? 
  7. Page 5, line 132: DNP or DPN? 
  8. Page 9, line 231: figure reference looks incorrect. Should it be figure 5? 
  9. Page 9, line 236: figure reference looks incorrect. Should it be figure 3?
  10. Page 9, Disruption of Schwann cells: this section talks about how Schwann cells are involved in DPN which we all know. Need more reference showing how Schwann cells pathology leads to compromised BNB function in diabetes studies (animals or human). 
  11. Page 9, Etiology of conditions other than hyperglycemia: This section needs references showing how factors other than hyperglycemia contribute to BNB disruption. Otherwise, it is irrelevant to mention the difference of etiology between type 1 and type 2 diabetes. And is it possible that BNB disruption is more likely in type 2 diabetes than type 1 due to involvement of additional factors beyond hyperglycemia? 
  12.  Page 10: Conclusions: lines 283-293 are exact copy paste as in introduction. It should be phrased differently.
  13. Same repetition for figure 6 legend (lines 296-305) is the same text again. This need to be simplified.

Author Response

Reviewer 1

General comments

  1. Introduction need to be better phrased with more detailed (one paragraph) about what happens to BNB in healthy and diseased/diabetic conditions.

Reply: In this article, we focus the BNB pathology in DPN and in vitro BNB model. To clarify these subjects, we rewrote the introduction.

  1. Conclusion was almost same as introduction which need to stand alone, phrased differently and with more future perspectives for a review article.

Reply: We rewrote the conclusion.

  1. English need to be improved in many places for clarity and to avoid misleading information. 

Reply: Before resubmitting, this article was checked by native speaker.

Specific comments:

  1. Abstract- Page 1, line 9: the sentence says "axons of Schwann cells" which is bizarre, how can Schwann cells have axons?

Reply: We regret this error which has been corrected.

  1. Introduction- Page 1, line 34-35: This reference is incorrect and old. Need a more recent reference (e.g.; PMID: 31518657)

Reply: We exchanged it for recent reference.

  1. Page 2, line 45-46: the sentence says: ..."peripheral nerve tissues, such as retina and kidneys"  retina and kidneys are not peripheral nerve tissues. 

Reply: We apologize and endeavored to correct the grammer comprehensively.

  1. References are missing in several places: Page 2, line 47-49 (In a hyper....  tissues). Page 2, line 61-63 (Like the ......   peripheral nerves). Page 2, end of line 68. Page 9, line 257-258 (Owing to ...   nerve tissues). Page 9, line 28-261 (When ......are induced). Page 9, line 264-266 (In a hyperglycemic.. CD*+ T-cells).

Reply: We regret this error which has been corrected.

  1. Figure 1: Title should be " comparative anatomy of BBB and BNB"

Reply: We agree with this suggestion.

  1. Figure 2: fix English in title. Hard to distinguish panel A and B. Is there a colored image? What was the scale bar? 

Reply: We regret this error which has been corrected.

  1. Page 5, line 132: DNP or DPN? 
  2. Reply: We regret this error which has been corrected.
  3. Page 9, line 231: figure reference looks incorrect. Should it be figure 5? 

Reply: We regret this error which has been corrected.

  1. Page 9, line 236: figure reference looks incorrect. Should it be figure 3?

Reply: We regret this error which has been corrected.

  1. Page 9, Disruption of Schwann cells: this section talks about how Schwann cells are involved in DPN which we all know. Need more reference showing how Schwann cells pathology leads to compromised BNB function in diabetes studies (animals or human). 

Reply: The mechanism of disruption of Schwann cells in DPN are enormous.Such a preparation is beyond the scope of the present study. We exchanged this chapter to “The association between disruption of Schwann cells and BNB”

  1. Page 9, Etiology of conditions other than hyperglycemia: This section needs references showing how factors other than hyperglycemia contribute to BNB disruption. Otherwise, it is irrelevant to mention the difference of etiology between type 1 and type 2 diabetes. And is it possible that BNB disruption is more likely in type 2 diabetes than type 1 due to involvement of additional factors beyond hyperglycemia? 

Reply: We agree with this opinion. Unfortunately, the molecular mechanisms of these risk factors in BNB dysfunction are still unknown.

  1.  Page 10: Conclusions: lines 283-293 are exact copy paste as in introduction. It should be phrased differently.

Reply: We regret this error which has been corrected.

  1. Same repetition for figure 6 legend (lines 296-305) is the same text again. This need to be simplified.

Reply: We regret this error which has been corrected.

Reviewer 2 Report

This review is a reasonably interesting introduction of pathogenesis of diabetic neuropathy, with special attention to the blood-nerve barrier dysfunction. Although the content contains not much novelty to the field (with only 7 out of 61 references published in the last 5 years), it is well summarized. There are several issues that could be incorporated for the improvement of the text and for the avoidance of misinterpretation of the data. My major suggestions are:

Title

1) I would suggest that the authors change the title, for example to something like “Blood-nerve barrier pathology in diabetic peripheral neuropathy”

Abstract:

2) Blood-nerve barrier, axons, Schwann cells and neurons are not peripheral nerve tissues but nerve components. Please re-phrase.

3) I don’t think the statement that hyperglycemia is responsible for the metabolic disorder in the peripheral nerve is totally correct. As the authors mention later in the manuscript, there are other important factor contributing to the pathology, e.g. dyslipidemia. In fact, it is known that e.g. Schwann cells are more sensitive to dyslipidemia than high glucose levels, so this needs re-phrasing, as it is not only the high sugar levels that leads to diabetes.

4) The expression “we have been successful, for the first time in the world…” needs re-phrasing as it doesn’t sound good. I would suggest something as “We have developed, for the first time,…”

5) The abstract is, in general, well written but it is not very obvious what this review is really about. I would recommend adding a goal for the manuscript and perhaps shorten a bit the introduction section of the abstract. Obviously (from the title) the reader knows that the paper will be about the BNB in diabetic neuropathy, but what is the authors main message and main conclusions with this work?

Introduction:

6) I understand that the authors want to highlight the diabetes numbers in their country, but since this is a publication for an international journal, I would remove the sentence in line 38/39 about Japan.

Anatomy of the BNB and peripheral nerves:

7) In line 57, I understand that BNB functions as a barrier mechanism, but how do Schwann cells and axons work here in terms of barrier? This needs re-phrasing.

8) Is it really true that BNB is as robust as BBB? Because in BBB there is the astrocyte extra layer and I believe that BNB is leakier as compared to BBB. So please confirm these facts and provide proper references.

9) Legend to figure 1 is incorrect. This is not the anatomy of a peripheral nerve. These sketches represent the BBB, BNB and a myelinating Schwann cell engulfing an axon with myelin layers.

10) Some deeper description of the temperature-sensitive SV40 large T-antigen gene methodology is necessary for the reader understanding. Are the cells initially cultured at 33 degrees? How was differentiation of endothelial cells different from that for pericytes? It is necessary to explore deeply why this technique can be relevant for future research and how it can add advantages to current cell lines for being used for pathology investigation of the BNB.

Are there any molecular studies published using this technique that have contributed for the understanding of the BNB pathology (that are not from the authors)?

BNB disruption

11) Letter formatting in legend of Figure 3 is wrong (lines 161 to 163). English needs some re-phrasing.

12) Authors focus the present review on hyperglycemia. What is the role of dyslipidemia in BNB disruption?

13) Regarding the main pathological and morphological changes of BNB in diabetic neuropathy, there is actually a recent review by Richner at al., Front Neurosci, 2019 that should be included as a reference.

14) There is an error in legend of Figure 4. It is indicated that the left panel is from a healthy individual and later in the legend it is written that C is a diabetic patient. Needs correction.

15) It is very interesting to find swollen endothelial cells. Is this a common pathological finding in nerve biopsies from diabetic patients with neuropathy? Can we say there is nucleus invasion of the capillary lumen? How do you envision that endothelial cell swollen takes place? Is it expanding? Is the basal membrane pushing the nucleus into the lumen?

16) Legend of Figure 5: I would add the word “modeling” after mechanism, for better understanding. English needs re-phrasing in general here.

17) Very recent knowledge was added into the role of inflammation for diabetic neuropathy by using type 2 diabetes mouse studies (Gonçalves et al., Glia, 2020).

18) Line 222 – reference 32 is not related with diabetic human patients.

19) Line 231 – I believe here authors refer to figure 4 and not figure 2? Same for line 236.

20) My overall feeling after reading this review is that it does not contain sufficient updated literature review. I would advise the authors to include more recent studies (published in the last 5 years) that might have contributed to the field with novel molecular insights. 

Author Response

This review is a reasonably interesting introduction of pathogenesis of diabetic neuropathy, with special attention to the blood-nerve barrier dysfunction. Although the content contains not much novelty to the field (with only 7 out of 61 references published in the last 5 years), it is well summarized. There are several issues that could be incorporated for the improvement of the text and for the avoidance of misinterpretation of the data. My major suggestions are:

Title

  • I would suggest that the authors change the title, for example to something like “Blood-nerve barrier pathology in diabetic peripheral neuropathy”

Reply: We agree with this suggestion. In this article, we focus the BNB pathology in DPN and in vitro BNB model. We exchanged this title to “Blood-nerve barrier (BNB) pathology in diabetic peripheral neuropathy and in vitro human BNB model.”

Abstract:

  • Blood-nerve barrier, axons, Schwann cells and neurons are not peripheral nerve tissues but nerve components. Please re-phrase.

Reply: We regret this error which has been corrected.

  • I don’t think the statement that hyperglycemia is responsible for the metabolic disorder in the peripheral nerve is totally correct. As the authors mention later in the manuscript, there are other important factor contributing to the pathology, e.g. dyslipidemia. In fact, it is known that e.g. Schwann cells are more sensitive to dyslipidemia than high glucose levels, so this needs re-phrasing, as it is not only the high sugar levels that leads to diabetes.

Reply: We agree with this opinion. As you know, Schwann cells are sensitive to dyslipidemia. Unfortunately, the molecular mechanisms of these risk factors in BNB dysfunction are still unknown. In abstract, we mention about it.

  • The expression “we have been successful, for the first time in the world…” needs re-phrasing as it doesn’t sound good. I would suggest something as “We have developed, for the first time,…”

Reply: We apologize and endeavored to correct the grammer comprehensively.

  • The abstract is, in general, well written but it is not very obvious what this review is really about. I would recommend adding a goal for the manuscript and perhaps shorten a bit the introduction section of the abstract. Obviously (from the title) the reader knows that the paper will be about the BNB in diabetic neuropathy, but what is the authors main message and main conclusions with this work?

Reply: In this article, we focus the BNB pathology in DPN and in vitro BNB model. To clarify these subjects, we rewrote the abstract.

Introduction:

 I understand that the authors want to highlight the diabetes numbers in their country, but since this is a publication for an international journal, I would remove the sentence in line 38/39 about Japan.

Reply: We agree with this opinion. This sentence was deleted.

Anatomy of the BNB and peripheral nerves:

  • In line 57, I understand that BNB functions as a barrier mechanism, but how do Schwann cells and axons work here in terms of barrier? This needs re-phrasing.

Reply: We apologize and endeavored to correct the grammer comprehensively.

  • Is it really true that BNB is as robust as BBB? Because in BBB there is the astrocyte extra layer and I believe that BNB is leakier as compared to BBB. So please confirm these facts and provide proper references.

Reply: This question is important. Although it is well-known that BNB have almost the same properties as a barrier system as those with BBB in BNB reaserch field, nobody knows the reasons. I added other reference #8 and our hypothesis.

  • Legend to figure 1 is incorrect. This is not the anatomy of a peripheral nerve. These sketches represent the BBB, BNB and a myelinating Schwann cell engulfing an axon with myelin layers.

Reply: We agree with this opinion. We exchange the title of this legend to “Comparative anatomy of blood-brain barrier and blood-nerve barrier”

  • Some deeper description of the temperature-sensitive SV40 large T-antigen gene methodology is necessary for the reader understanding. Are the cells initially cultured at 33 degrees? How was differentiation of endothelial cells different from that for pericytes? It is necessary to explore deeply why this technique can be relevant for future research and how it can add advantages to current cell lines for being used for pathology investigation of the BNB.

Reply: When transfected cells with temperature-sensitive SV40 large T-antigen are cultured at 33 degrees, their growth speeds were almost twice as slow as HEK293 and transfected cells with SV40 large T-antigen. At 37 degrees, they can be alive only for 5 days. But for the term their barrier function increased 1.5 time and the expression of tight junction molecules increased. Although the morphology of transfected cells with SV40 large T-antigen is like cobble-stone, transfected cells with temperature-sensitive SV40 large T-antigen can keep the spindle-shape. In these points, transfected cells with temperature-sensitive SV40 large T-antigen have better BNB properties than SV40 large T-antigen.

Are there any molecular studies published using this technique that have contributed for the understanding of the BNB pathology (that are not from the authors)?

Reply: Please check reference #15.

BNB disruption

  • Letter formatting in legend of Figure 3 is wrong (lines 161 to 163). English needs some re-phrasing.

Reply: We regret this error which has been corrected.

  • Authors focus the present review on hyperglycemia. What is the role of dyslipidemia in BNB disruption?

Reply: The role of dyslipidemia in BNB disruption is not clarified.

  • Regarding the main pathological and morphological changes of BNB in diabetic neuropathy, there is actually a recent review by Richner at al., Front Neurosci, 2019 that should be included as a reference.

Reply: We agree with this opinion. We added this review in reference #35.

  • There is an error in legend of Figure 4. It is indicated that the left panel is from a healthy individual and later in the legend it is written that C is a diabetic patient. Needs correction.

Reply: We regret this error which has been corrected.

  • It is very interesting to find swollen endothelial cells. Is this a common pathological finding in nerve biopsies from diabetic patients with neuropathy? Can we say there is nucleus invasion of the capillary lumen? How do you envision that endothelial cell swollen takes place? Is it expanding? Is the basal membrane pushing the nucleus into the lumen?

Reply: Swollen endothelial cells are common in DPN. The nucleus of endothelium doesn’t invade the lumen. We think hyperglycemia failure the aquaporin channel of endothelilum.  It is possible that thicken basal membrane push the nucleus into the lumen.

  • Legend of Figure 5: I would add the word “modeling” after mechanism, for better understanding. English needs re-phrasing in general here.

Reply: We agree with this opinion.

  • Very recent knowledge was added into the role of inflammation for diabetic neuropathy by using type 2 diabetes mouse studies (Gonçalves et al., Glia, 2020).

Reply: We added this in reference #44

  • Line 222 – reference 32 is not related with diabetic human patients.

Reply: We regret this error which has been corrected.

  • Line 231 – I believe here authors refer to figure 4 and not figure 2? Same for line 236.

Reply: We regret this error which has been corrected.

20) My overall feeling after reading this review is that it does not contain sufficient updated literature review. I would advise the authors to include more recent studies (published in the last 5 years) that might have contributed to the field with novel molecular insights. 

Reply: We added five recent studies in this review and mentioned the molecular mechanisms with them.

Round 2

Reviewer 2 Report

The authors have successfully replied to all my comments.

I know recommend this review manuscript for publication.